# Facilitating Sustainable Disaster Risk Reduction in Indigenous Communities: Reviving Indigenous Worldviews, Knowledge and Practices through Two-Way Partnering

**DOI:** 10.3390/ijerph18030855

**Published:** 2021-01-20

**Authors:** Tahir Ali, Petra Topaz Buergelt, Douglas Paton, James Arnold Smith, Elaine Lawurrpa Maypilama, Dorothy Yuŋgirrŋa, Stephen Dhamarrandji, Rosemary Gundjarranbuy

**Affiliations:** 1College of Health and Human Sciences, Charles Darwin University, Darwin 0810, Australia; douglas.paton@cdu.edu.au; 2Faculty of Health, University of Canberra, Canberra 2617, Australia; petra.buergelt@canberra.edu.au; 3Adjunct Professor, Faculty of Health, University of Canberra, Canberra 2617, Australia; 4Wellbeing and Preventable Chronic Conditions, Menzies School of Health Research, Darwin 0810, Australia; james.smith@menzies.edu.au; 5College of Indigenous Futures, Arts & Society, Charles Darwin University, Darwin 0810, Australia; Elaine.maypilama@cdu.edu.au; 6Yalu Marŋgithinyaraw, Galiwin’ku, East Arnhem Land, The Northern Territory 0822, Australia; Dorothy.yungirrja@yalu.org.au (D.Y.); stevenlot133@gmail.com (S.D.); Rosemary.Gundjarranbuy@yalu.org.au (R.G.)

**Keywords:** Indigenous development, holistic Indigenous research, disaster risk reduction, sustainable community development, partnership-based approaches, critical Indigenous methodology

## Abstract

The Sendai Framework of Action 2015–2030 calls for holistic Indigenous disaster risk reduction (DRR) research. Responding to this call, we synergized a holistic philosophical framework (comprising ecological systems theory, symbolic interactionism, and intersectionality) and social constructionist grounded theory and ethnography within a critical Indigenous research paradigm as a methodology for exploring how diverse individual and contextual factors influence DRR in a remote Indigenous community called Galiwinku, in the Northern Territory of Australia. Working together, Indigenous and non-Indigenous researchers collected stories in local languages using conversations and yarning circles with 20 community members, as well as participant observations. The stories were interpreted and analysed using social constructivist grounded theory analysis techniques. The findings were dialogued with over 50 community members. The findings deeply resonated with the community members, validating the trustworthiness and relevance of the findings. The grounded theory that emerged identified two themes. First, local Indigenous knowledge and practices strengthen Indigenous people and reduce the risks posed by natural hazards. More specifically, deep reciprocal relationships with country and ecological knowledge, strong kinship relations, Elder’s wisdom and authority, women and men sharing power, and faith in a supreme power/God and Indigenous-led community organizations enable DRR. Second, colonizing practices weaken Indigenous people and increase the risks from natural hazards. Therefore, colonization, the imposition of Western culture, the government application of top-down approaches, infiltration in Indigenous governance systems, the use of fly-in/fly-out workers, scarcity of employment, restrictions on technical and higher education opportunities, and overcrowded housing that is culturally and climatically unsuitable undermine the DRR capability. Based on the findings, we propose a Community-Based DRR theory which proposes that facilitating sustainable Indigenous DRR in Australian Indigenous communities requires Indigenous and non-Indigenous partners to genuinely work together in two-directional and complementary ways.

## 1. Introduction

Indigenous peoples have, over several millennia, developed sophisticated ecological knowledge and practices to predict, prepare for, cope with, and survive natural events using their close intimate relationships with the country in which they live [1,2]. However, Australia’s colonizers disregarded the Indigenous disaster risk reduction (DRR) perspectives, knowledge, and capacities that had developed, dismissing them as either primitive or irrelevant. Instead, they imposed Western risk management systems on Australian Indigenous peoples (Australian Indigenous peoples include Aboriginal and Torres Strait Islanders. However, Aboriginal and Torres Strait Islander peoples differ in many ways, including in terms of their identity, culture, language, beliefs, traditions, protocols, issues, and history in pre- and post-colonial contexts [3]. Because our research was conducted with the Yolŋu community in Galiwin’ku, who are Australian Indigenous peoples, we refer to Indigenous peoples in this paper) [4,5,6,7]. These top-down, hazard-focused, deficit-based, response-oriented, decision-centralized, and agency-driven systems overrode and undermined Indigenous DRR capacities [1,8,9]. The fact that such approaches have failed in Indigenous settings and in broader global contexts [5,10] highlights the urgent need to transform the worldviews underlying traditional and contemporary Western DRR in ways that recognize Indigenous worldviews and utilize Indigenous knowledge and practices in community-based DRR [1]. This is not to say that Western knowledge is not applicable. However, its relevance must be understood as an adjunct to traditional knowledge, rather than a replacement for it. Enacting the latter approach calls for shifts in thinking among Western-based DRR researchers.

One way of facilitating this paradigm shift is to conduct genuinely collaborative research partnerships involving Indigenous peoples and their Western counterparts in ways that facilitate Indigenous peoples reviving and strengthening their Indigenous worldviews, knowledge, and practices and applying them to DRR [11,12]. Such two-way or reciprocal approaches must involve greater coordination between Indigenous and non-Indigenous stakeholders, in order to understand their respective DRR perspectives and integrate the best of both sets of knowledge to support sustainable development at local, national, and global levels [7,8,11].

To address these calls and contribute to facilitating DRR in Australian Indigenous communities, we conducted a qualitative study with a remote Indigenous community, known as Galiwinku, in the Northern Territory of Australia (Figure 1). The goal was to develop a holistic and all-hazards Indigenous community-based DRR theory grounded in the lived experiences of Indigenous peoples that reflects their perspectives on how colonialization impacted their DRR capability; what needs to be done to counter this; and what intrinsic beliefs, knowledge, relationships, and practices must be developed to support a sustainable, Indigenous community-based DRR capability. We sought to achieve this by seeking answers to the following research questions: (i) What historical and contemporary individual and contextual factors and processes interact over time to influence DRR and community development in Indigenous communities? (ii) How do Indigenous knowledge systems, worldviews, and practices strengthen individual and collective coping capacities to reduce disaster risks? (iii) How can Indigenous coping capacities be facilitated, and the undermining factors be mitigated, in Indigenous communities?

We now present a brief overview of the literature highlighting the social and environmental risks faced by remote Indigenous communities in Northern Australia. This section also discusses current DRR research and practice, and the problems it creates, with specific reference to Indigenous development. The subsequent sections explain our research design and findings and their implications for DRR in remote Indigenous contexts in Australia.

### 1.1. Disaster Risks of the North Australian Remote Indigenous Communities

Almost 100,000 Indigenous people in the Northern Territory (NT), Australia (about 50% of the total NT population) live in remote tropical coastal communities [4,11]. These communities face considerable risks from natural hazards, especially those emanating from climate change [5,13]. These challenges are compounded by the sea level rise of 18 cm that has occurred in the NT over the last 20 years (twice the global average) and the severity of category 3–5 cyclones (hurricanes) increasing by 60% [14]. These sea-level and cyclone risks interact with the low topography of the NT coast, resulting in cyclone and flood hazards reaching several kilometers inland and affecting a substantial number of remote Indigenous communities [15]. Furthermore, projected increases in the average temperature of 1.5 degrees Celsius in the NT by 2030 [14] will add to extensive and intense bushfires (wildfires), causing extreme heat and humidity within areas occupied by remote Indigenous communities [16,17].

Structural and infrastructure problems further aggravate the environmental risk faced by rural Indigenous communities of the NT. For example, construction and communication service issues, overcrowding, and inadequate housing (disrepair, limited availability, size and adequacy, location, and lack of safety codes and standards) represent built environmental factors that contribute to the risk [5,8,18]. Furthermore, demographic factors, such as low population densities and high population turnover, remoteness, and the growing use of fly-in/fly-out (FIFO) services, introduce social challenges to the disaster risk planning context in remote communities. Contributions to risk can also be traced to historical factors.

The devastating impacts of colonization place Australian Indigenous peoples among the highest at-risk communities in terms of environmental and health sources [6,9]. The social exclusion, discrimination, racial inequality, and partisan political agendas that colonizers introduced brought disempowerment, trauma, distress, and sickness to Australian Indigenous people [9,19]. Consequently, compared to their non-Indigenous counterparts, Australian Indigenous peoples experience a lower education rate (43% compared to 67%), a 16 times greater mortality, and a 3 times higher suicide rate [9]. Additionally, they are twice as likely to be exposed to violence and 13 times more likely to be imprisoned [9].

Colonization had additional implications for the fundamentally important ways that Australian Indigenous peoples relate to their “country” (Australian Indigenous peoples refer to the land as country) and environment. Country is central to Indigenous peoples, both individually and collectively [1]. Indigenous people derive their identity, health and well-being, livelihoods, and cultural and spiritual capacities from their close, reciprocal, country-specific, and knowledge-based connections to country and peoples [9,19,20]. However, the most detrimental legacy of colonization beliefs and practices in Australian Indigenous lives is the dispossession of country that continues to date at local and national levels [7,9,19]. These injustices, in turn, eroded the socio-cultural-environmental adaptive capacities that Indigenous peoples used as risk management and response mechanisms, including Indigenous worldviews, ecological knowledge, intimate reciprocal relationships with nature, bonds within and between clans, and spiritual and cultural practices [1,8]. These practices are increasingly being recognized as important for DRR [1,2,7,8,13]. Consequently, they must be included in the development, implementation, and evaluation of DRR policies and practices.

However, pursuing this goal involves more than simply recognizing the value of these practices. It is essential to first research and understand how historical and contemporary colonization practices have undermined, and continue to systematically undermine, the adaptive capacities that Indigenous peoples can draw on to play pivotal roles in DRR, response, and recovery practices. In other words, the damaging influence of colonial beliefs and practices must be understood, and their influence be addressed if a goal of developing sustained Indigenous DRR practices is to be realized. How this goal can be pursued is discussed next as we explore how historical and contemporary colonial processes have been understood and accounted for in Indigenous DRR research and policy formulation and implementation in Northern Australia.

### 1.2. Current Indigenous DRR Research and Practice

The broad spectrum of interrelated factors (e.g., social, environmental, cultural, etc.) that influence Indigenous risk and the DRR strategies required imply that a holistic and in-depth approach is required to gain a comprehensive understanding of both how environmental hazards impact people and the beliefs, knowledge, relationships, and practices required to facilitate DRR and disaster recovery in Indigenous communities. A foundational issue here is that the current DRR research discourse applied in this context is predominantly aligned with a reductionist Western paradigm that focuses on hazard and risk assessment. Advancing the development of an Indigenous DRR calls for a different approach, and one that focuses on understanding how to adapt Indigenous beliefs and strategies to the diverse and context-specific consequences that people experience. Building on this knowledge, it then becomes possible to develop response and adaptive approaches that must be applied to facilitate the development of sustainable approaches to reducing the target population’s risk and increasing their response capability [21,22].

In practice, the reductionist research approaches introduced above have led to the formulation of impoverished development policies that do not integrate community development with risk reduction to sustainably address the underlying reasons for the disaster risks, health, and quality of Indigenous peoples’ lives [12]. For example, in Northern Australia, emergency management still relies on the traditional disaster management cycle, with four isolated stages of mitigation, preparedness, response, and recovery [2,6]. Such approaches focus on technological warnings, expensive evacuations, and rebuilding infrastructure [6,23], rather than developing more cost-effective, culturally appropriate community-level approaches to manage risks and facilitate response capability. Furthermore, the application of top-down emergency management models generally fails to utilize contemporary academic and local knowledge to inform DRR and recovery planning in Indigenous settings [24]. Similarly, development policies applied in remote Indigenous communities in the NT continue to address the economic vulnerabilities that contribute to natural hazard risks in ways that that are independent of DRR policy and practice [6,13]. Collectively, these practices combine to create costly and ineffective DRR capabilities and fail to capitalize on development opportunities across policy and practice platforms to create more community-focused DRR capability.

Consequently, a significant research gap thus derives from challenges associated with exploring and understanding the interdependent contextual social, cultural, and environmental factors that interact over time to increase the likelihood of natural events turning into disasters. Understanding the latter requires accommodating the long-term structural adjustments required to empower people and communities. This includes integrating, for example, the employment, health, and education factors that make implicit contributions to risk within a DRR framework [25,26]. Past failures to accommodate these factors meant that top-down, Western DRR research and practices failed to reduce social and community risks because they focus on DRR components in isolation from the local social realities, perspectives, needs, and requirements in a holistic way that underpin the community capacity [10,27]. Recognition of the need to accommodate the latter when conceptualizing and delivering DRR strategies has drawn attention to the value of including community development practices in DRR thinking.

The value of a community development approach derives from the fact that a significant outcome of applying top-down (colonial) policy and practices is a lack of appreciation of the crucial relationships and interdependencies between the people and their natural and socio-ecological environment that represent the bedrock of cultural life and that are crucial to effective DRR [28]. By failing to adopt such an approach, prevailing top-down governance practices have led to fragmented communities and social DRR policies that increase the environmental risk that communities are exposed to [29,30]. Furthermore, existing research, policies, and practices developed within this Western paradigm seldom empower remote Indigenous communities to (re)develop long-term and sustainable self-reliant adaptive capacities that (re)enable them to proactively reduce the risks posed by extreme natural environmental events and utilize their knowledge to effectively respond to and recover from disasters that do occur [31]. We argue, as others have, that a more holistic, community development-based approach is required to integrate research across domains involving transdisciplinary system analysis of DRR [21,25,29,32,33].

The holistic approach we adopt aligns well with the Indigenous philosophy of holism, which focuses on social, emotional, and cultural well-being in ways that holistically facilitate a “whole of community” approach to achieving people’s (individual and collective) full potential [34]. This Indigenous philosophy underscores the call from all of the major DRR and community development frameworks used in Australia, including Closing the Gap [35], the Council of Australian Government Report [36], Keeping Our Mob Safe [37], and the National Strategy for Disaster Resilience [38], for holistic Indigenous research and policy formulation. We contend that our study thus represents a timely and much-needed response to this call.

## 2. Methodology

This study drew upon three intersecting philosophical frameworks, including Ecological Systems Theory [39], Symbolic Interactionism [40], and Intersectionality [41]. The intersections between these three frameworks facilitate critical investigations of Indigenous DRR as a whole system and explorations of Indigenous people’s lived experiences, interpretations, and actions/interactions from their perspective. Ecological Systems Theory guided us to systematically identify the individual and contextual factors and processes at different levels of the DRR system and how they interact historically and contemporarily to influence interpretations and actions/interactions over time. Symbolic Interactionism facilitates our understanding of how Indigenous peoples attach meaning to and interpret extreme natural events and their relationship with DDR processes and practices, and how these interpretations influence their actions/interactions at the various system levels. Intersectionality directed our attention to specifically and critically investigating, and making explicit, the influence of discriminatory and deficit-based colonial practices on Indigenous DRR.

The research was conducted with the Galiwin’ku community on Elcho Island in the Northern Territory of Australia (Figure 1). Galiwin’ku is a coastal island of Arnhem Land about 550 kilometers northeast of Darwin. The community is surrounded by the Arafura Sea on the West and the Cadell Strait on the East. Galiwin’ku connects with Darwin and surrounding communities through air flights (mainly small planes). It has a population of about 2200 people, 97% of whom are local Indigenous peoples who identify as Yolŋu [42].

Yolŋu belong to two main moieties called *Dhuwa* and *Yirritja* [43]. According to Yolŋu philosophy, everything in the universe belongs to either moiety. A main source of connection between the two moieties is marriage, as marriage within a moiety is not allowed [43]. In this sense, the husband belongs to one moiety and the wife belongs to the other moiety. Children belong to their father’s moiety. Within each moiety, people belong to smaller groups called clans. There are around sixteen clans in Galiwinku.

Galiwin’ku was selected for the research because it experienced two back-to-back category 4 cyclones, named Lam and Nathan, in February and March 2015, respectively. Most of the island’s infrastructure, including 80 houses, was destroyed or severely damaged in these cyclones [44]. Given the recency of these events, community members were able to provide detailed recollections of their experiences, interpretations, and actions/interactions.

To ensure that the research was conducted in a culturally suitable way [45], and to ensure that data reflect the lived experiences and understandings of Indigenous people [46], the research design was developed in conjunction with the Indigenous PhD supervisor (L.M.) and the non-Indigenous supervisors, who have extensive experience conducting research with Indigenous peoples (P.B., D.P., and J.S.). L.M. is a Yolŋu elder who belongs to and lives in Galiwin’ku and who is a senior researcher with nearly 40 years of experience conducting research with Indigenous communities in Arnhem Land.

To gain an understanding of how the lived DRR experiences, interpretations, and actions/interaction of Yolŋu interact within and across different levels to influence DRR, we merged a case study approach [47], social constructionist grounded theory [48], and ethnography [49] within an Indigenous research paradigm [45] as a methodology to guide the research process. Data were collected through conversations [50], yarning circles [51], and participant observations [49]. These methods are recognized as culturally appropriate for conducting research with Indigenous peoples [49,50].

In line with constructionist grounded theory, we utilized purposeful and theoretical sampling to ensure that diverse perspectives were obtained. To include the voices of a wide range of Indigenous peoples, twenty Yolŋu participants with diverse relevant backgrounds were invited to participate. This included Traditional Owners, Yalu Marŋgithinyaraw staff (Yalu is a community-based research organization), board members of the Shire (Local Authority), schoolteachers, a member of the local police, and community members (Table 1). All of the participants, except one, consented to the use of their real names, rather than pseudonyms. An important aspect of decolonizing Western research and upholding Indigenous data sovereignty is to respect that some Indigenous participants consider it very important to be identified with their real names, in order to show respect towards their relationships and to indicate their authority to share and present their information [50,52]. To ensure this, in the consent form, we decided to give participants the option to choose either their real names or pseudonyms. Approval for participants being able to choose their real names and using participants’ real names when employing quotes from them in publications was obtained via ethics applications from Charles Darwin University, Australia, and the University of Canberra, Australia, under clearance number H18060.

The data were collected in February 2020. To collect the data, the first author T.A. lived in Galiwin’ku for three weeks. Two local Yolŋu co-researchers, referred to as D.Y. and S.D., from Yalu were selected and employed via Yalu to conduct the data collection, interpretation, and analysis. They, together with L.M. and T.A., completed the write-up of the data and findings. The Indigenous supervisor L.M. provided training for T.A. and the local Indigenous co-researchers to develop a shared and culturally embedded approach to understanding the purpose of the research, identifying relevant topics, and adopting culturally appropriate ways to explore people’s experiences and interpretations [53]. The other supervisors, especially P.B., co-implemented the project with L.M., T.A., and the community-based researchers.

To ensure responsibility and accountability for the research process and reporting [45,50,52,53], a partnership was established with Yalu. This partnership was formally acknowledged with a memorandum of understanding between Yalu and the University of Canberra. The University of Canberra provided funding for this research via its Collaborative Indigenous Research Initiative. Yalu acted as a Steering Committee to ensure that an ethical and sensitive research process was adopted and applied.

Conversational data were collected through one-to-one conversations or yarning circles, depending on the preference of the participants. Participants spoke in the local language or in English during the conversations, as per their choice. Details of the participants, the type of participation, and the number and duration of conversations and yarning circles are shown in Table 1. To gain insights into the actions/interactions as they occur in the daily lives of Yolŋu, T.A. observed and participated in daily activities, such as sharing meals, shopping, fishing, hunting, and playing sports. These data were recorded in the form of field notes by T.A. in a daily journal.

The data were analysed using constructivist grounded theory analysis techniques developed by Charmaz [48], with the help of Atlas.ti. Atlas.ti is a computer-assisted qualitative data analysis software which is particularly helpful in inductively and non-hierarchically analysing text-based documents [54].

In September 2020, the research team engaged in two-way feedback sessions conducted over a five-day period. During this time, the findings were dialogued with over 50 Yolŋu through the Local Authority (comprising the Traditional Owners and non-Indigenous Shire Representatives), Yalu, and seven community groups. The collective story that emerged from the research deeply resonated with the community members, validating the research and analysis. Through this process, community members extended the scope of the analysis, and their feedback on cultural, historical, and contextual issues was incorporated into the analysis.

## 3. Findings

From the analysis, emergent codes were grouped into 16 sub-categories (blue boxes in Figure 2). The sub-categories were merged into two overarching categories (yellow boxes). The categories were (i) Yolŋu’s knowledge and practices strengthening Yolŋu, reducing the risk of extreme natural events and disasters, and (ii) colonizing practices weakening Yolŋu and increasing the risk of natural events becoming disasters. The analysis of categories and their relationship with each other yielded a core category or process (red box), which was “Yolŋu reclaiming power by reviving and strengthening Indigenous knowledge through two-way learning with non-Indigenous partners”. The core category implies that, to be effective, DRR in Galiwin’ku needs to support both processes: Yolŋu reviving and strengthening their worldviews, knowledge, and practices, as well as working together in a two-way approach and co-creating and co-implementing DRR. Next, we will present the findings in detail according to the sub-categories and categories identified, starting with the “old days” (Figure 2).

### 3.1. Yolŋu Knowledge and Practices Strengthen Yolŋu, Reducing the Risk of Natural Events Becoming Disasters

#### 3.1.1. We Are the Land: The Land Is Us

The Yolŋu worldview involves deep reciprocal relationships with country: *We are the land and the land are us* (Tamara). These connections are regarded as important determinants of the healthy continuity of all aspects of life and thus the health of people and country. Participants argued that, with their traditional reciprocal connections with the country, there were no disturbances in nature. They stated that, because of people’s efforts to live harmoniously (co-exist) with nature, cyclones used to be of a low intensity, and seldom caused damage. Consequently, cyclones were never seen as a threat or potential source of disaster. Rather, they were perceived as the recycling of life; a normal natural process that cleaned and renewed the country, as well as prevented extreme natural events and disasters: 


*“A natural event is a natural event. Back in the old days, if a cyclone came in, slashed everything, it was a normal thing. It prevented the natural disaster itself. It is the recycle of the life”*
(Tamara).

In the context of such connected and harmonious social–environmental relationships, cyclones are depicted as part of normal life in Yolŋu Dreaming Stories, which are sung and cherished: *“The sharing of the Dreaming Stories, songs, ceremonies, art, language and history explaining Yolŋu connections with the environment make people strong”* (Maypilama). These worldviews were a strong source of the value of living in harmony with natural events, especially cyclones embodied in people’s worldview.

However, the imposition of the Balanda (Western) worldview and culture following the colonial invasion was identified as being responsible for triggering environmental disturbances and events which Yolŋu perceive as creating circumstances that exceed their traditional adaptive capabilities. Participants termed the Balanda culture and its consequences as the real disasters for Yolŋu: *“Now a days with balanda culture, the natural disasters have become much bigger. For example, sea full of plastic and rubbish”* (Tamara). The evolution of more intense cyclones, compared to the smaller ones experienced in the past, have intersected with people’s conception of cyclones as being the source of several (novel) negative consequences that need to be addressed by DRR.

Given their prevailing historical experience of small, low-intensity cyclones, people found it difficult to understand the reason for the occurrence of the more recent high-intensity cyclones Lam and Nathan (hereafter referred to as Cyclones) and the damage, trauma, and stress they caused: *“We felt weak and we were surprised why is this happening, what is causing this, who’s gonna help us”* (Dorothy). Such conceptions resulted in several dangerous practices during the Cyclones. For example, Galikali and Rosemary shared that, during the Cyclones, when everybody was meant to be in shelter or houses, some youngsters were roaming around, vandalizing and breaking into the shops. Galikali argued that this behavior was due to the mixed emotions of traditional excitement and fearing the Cyclones.

The deep emotional connections to country were severely hurt because of the damage caused by the Cyclones. This was particularly true regarding people experiencing the land being “crushed down and just broken into pieces”; this was heartbreaking, upsetting, and hurtful for Yolŋu due to them being intimately connected to this land and because they live on and from the land:


*“The native here, the trees, lands were just broken into pieces. It just drained me inside out”*
(Galikali).


*“There were no birds flying around, all our grass was brown and died in one day. It was not normal it gave us fear”*
(Djanice).

The losses from the Cyclones had a big impact on peoples’ psychological resilience. Everything being destroyed resulted in people feeling shocked, and feelings of being removed to some other place, sad, drained, hopeless, and depressed. They wondered the following:


*“What are we gona do here, how we gona rebuild our houses and support each other? How we gona build this community again?”*
(Galikali).

People’s deep emotional connections to country prompted a strong (collective) desire to participate in the community recovery and regeneration processes after the Cyclones. Participants believed that, by being local people who belong to these lands, their involvement in recovery and regeneration activities would add considerable value to the community development process. In contrast, they did not believe that this value could be brought to the community by the fly-in/fly-out (FIFO) staff with their Western response to the recovery: *“They (Balanda) do not work with hearts for us, but only with their minds”* (Rosemary).

#### 3.1.2. Our Knowledge Enables Understanding of Nature’s Cyclone Warning Signs

As shown in Figure 2, the deep intimacy with country, which spans over millennia, has resulted in the development of extensive and deep ecological knowledge. This knowledge has been passed from the old days (before colonization) to the present, from generation to generation. The extensive ecological knowledge is, for example, reflected in a calendar of the seasons Tamara’s grandmother developed based on the extensive ecological knowledge passed on to her. This calendar depicts seven different seasons of the year, including *Wolmay* (build-up), *Dhuludur* (pre-wet), *Mayaltha* (wet), *Gunmul* (mid-wet), *Midawarr* (end of wet), *Dharratharra* (cool-dry), and *Rarranhdharr* (hot-dry) seasons. These seasons correspond to nuanced changes in animals, plants, winds, and rain patterns, and the activities that need to be undertaken during each season. This cultural calendar has won several national awards.

This intimate knowledge of nature enables them to understand and use cues from observed changes in their environment to predict cyclones, often ahead of the release of agencies’ “early” warnings. All of the participants talked about how variations in animal and bird behavior, wind, and wave patterns acted as warnings of an impending natural event:


*“We noticed that the sky was still, the wind was still, the sea was calm, there were no birds flying. These were the signs of burmulala [cyclone]”*
(Rosemary).


*“If tides go out the wind goes slow. If tides go full and high, the wind goes harder and can bring cyclone. These are the continuous instructions we get from creation”*
(Shane).

Margaret shared the story of how she anticipated the Cyclone when she noticed the changes in her dog’s behavior, who became hyperactive and restless, two days before the Cyclone.

However, it emerged from the conversations that traditional signs of the Cyclone did not, as would have been the case in the past, necessarily persuade people to prepare for the imminent event (e.g., stockpiling food). This undermining of the function of traditional knowledge could be attributed to several factors derived from the growing imposition of colonial and Western practices. The most prominent was people’s dependence upon government agencies for information and preparedness. Accompanying this reliance, or deferring of responsibility, has meant that people have increasingly been ignoring both their cultural knowledge and practices regarding natural warnings and their traditional role in influencing when and how to prepare for cyclones. Participants also discussed the ever-increasingly dominant use of Western technological interventions, including TV, radio, and mobile phones, for hazard information. Reliance on the latter contributes to eroding the use of traditional (warning and preparedness) knowledge and practices. Furthermore, the imposition of technical knowledge through government sources affects the traditional patterns of social relationships that facilitate social aspects of preparedness. In contrast to the social contexts in which western DRR occurs, the Yolŋu setting is more complex and thus must be accommodated in DRR planning and actions.

#### 3.1.3. Yolŋu to Yolŋu Help Due to Caring and Sharing

There are sixteen different clans living together in Galiwin’ku. These clans are strongly connected through the kinship system, intermarriage, and ceremonies. These inter-connections have contributed to Yolŋu developing strong reciprocal relationships across the community. These relationships promote reciprocal caring and sharing, which greatly supports Galiwin’ku residents in both their everyday lives and in emergencies. It is these kinds of relationship systems that highlight the benefits of integrating community development processes in DRR and making greater use of family, tribal, and community social networks and their implicit functions and capabilities in preparedness, response, and recovery planning and strategies [55].

In the preparedness, response, and regeneration phases of the Cyclones, these close-knit relationships facilitated Yolŋu’s capabilities by extending their access to communal resources and social capacities. For example, in the preparedness phase, the strong relationships helped to spread information about the Cyclones from relevant agencies, including police and the local emergency management unit, to local community members speaking in local languages at the grassroots level: *“That was Yolŋu to Yolŋu help. We were spreading the news, sharing and talking, informing people that burrmalala is coming”* (Rosemary). The information from fellow community members with similar affinities, values, and interests helped people better understand cyclone risks and preparedness actions. What is clear is that understanding and accommodating traditional social relationships will be important when developing two-way or reciprocal research (and practice) relationships between Yolŋu and their Western counterparts. Accommodating social relationships is also pivotal to developing effective response capabilities.

In the response phase, for instance, strong relationships manifest among community members taking collective responsibility for supporting the most vulnerable members of the family, including children, elders, and the disabled: *“I chose to stay home to look after my mum, and her things”* (Shane). At the community level, relationships ensured that support was extended to vulnerable members of the community, including those with fewer financial and material resources and those less knowledgeable about the risks associated with cyclones. For example, neighbors helped other neighbors who faced problems moving to cyclone shelters because of the unavailability of public transport in the community. The values derived from the kinship system that create this extensive web of deep relationships imparted the sense of strong mutual obligation, shared responsibility, and purpose in Yolŋu, which led them to take an active role in helping their neighbors during the Cyclones:


*“I helped because of my relationships, they are part of me too, they are my people. I participated in going around, getting water, feeding the old people”*
(Nyomba).

In the slow government-managed recovery process, strong relationships were vital to defusing people’s trauma, providing functional social support (e.g., emotional, tangible, relationship, and belongingness support), and enhancing people’s psychological resilience. After the Cyclones in 2015, the government announced plans to rebuild damaged and collapsed houses. During this period, people lived in temporary housing known as ‘demountables’. Without consulting with Yolŋu, government agencies did not place these demountables in the areas where people previously resided. The government response thus displaced people and separated them from their families. The consequent prolonged displacements and separations caused great distress and trauma and created unnecessary challenges to accessing social support and collaborative approaches for residents managing their own recovery. Despite the inappropriate government practices implemented, strong family and community relationships ensured continuing communication among people and supported their collective use of cultural practices. These helped to restore the routine, in ways that contributed to sustaining the health and wellbeing of the people:


*“During our stay in demountables, people were visiting us. Sometimes we would sing together, some would go hunting and bring back food and share with everyone”*
(Djandi).

The emergence of the role of these collaborative practices in community recovery introduced other issues with a bearing on effective DRR. Amongst these, leadership and how this was distributed along gender lines were significant.

#### 3.1.4. Yolŋu Women and Men Sharing Power

In Yolŋu culture, women and men enjoy social equality; both coexist in different roles to ensure that sacred law is practiced and passed on to ensure that Yolŋu continue to live in harmony with nature and each other [1]. This places women and men in the roles of being custodians of certain parts of the law and knowledge. Women are custodians of kinship, marriage arrangements, and land relationships, and men are enforcers of the law within the context of cultural leadership roles [1]. The equal power of both genders has facilitated and ensured the perseverance, transmission, and implementation of traditional ecological knowledge in all aspects of life through the generations.

However, colonialism progressively eroded the role of men as law enforcers, with this being perceived as creating gaps in community leadership:


*“When we moved to the places with balanda [Western] services, the ḏirramu [men] somehow started losing their sense of responsibility. They started going to drink, playing cards, and stopped doing their jama (work)”*
(Dorothy).

The damage that colonial practices did to men’s leadership roles has hindered the implementation of Indigenous law and knowledge. This contributed to eroding, for example, the community’s ability to use ecological knowledge in DRR, predicting hazard events, and facilitating an effective response to hazard events that occur (see above). One consequence of this was that, when the Cyclones occurred, limitations in terms of Yolŋu elders being able to exercise leading roles before and after the Cyclones led to the women taking up leadership roles in the community:


*“There is one thing they [balanda] forgot to take. That was the power of miyalk [women]. Miyalk decided to come up, get strong and stand and take the role. We decided to lead and build ourselves”*
(Maypilama).

This was evident in Yolŋu women in Galiwin’ku taking visible roles in leading the community and being active in diverse activities designed to strengthen Yolŋu (e.g., health and wellbeing programs and research activities). Drawing upon their specific ancestral knowledge, wider relationships, and inherent nurturing nature, women in leadership roles have greatly facilitated community health, well-being, and development processes. For example, with their deeply embedded relationships, women were in better positions to identify the vulnerable individuals in the community who needed support during the Cyclones.

Nonetheless, the Western patriarchal worldviews and top-town disaster management approaches adopted by policy makers and disaster response agencies in Galiwin’ku meant that female leadership and knowledge-holder potential was never recognized and thus was not utilized in DRR. However, strengthening the traditional equality of both women and men would greatly support DRR and ensure gender-responsive DRR that accommodates the interests and concerns of both genders [56]. While the discussion in this section has introduced new ways in which leadership can be exercised, it also highlights a need to understand traditional wisdom and the authority systems that lay the foundations for effective DRR, and the governance processes that underpin its use.

#### 3.1.5. Yolŋu Elders’ Wisdom and Authority

As Figure 2 depicts, Yolŋu Elders are another vital local resource. Besides being the custodians of Yolŋu knowledge maintaining connected and harmonious relationships with nature and understanding the early warning signs that prevent disasters, Elders also facilitated the response to the Cyclones in Galiwin’ku. To mobilize the local resources and capacities during the Cyclones, government agencies approached the Elders. Being the custodians of Yolŋu knowledge bestows on Elders great respect, authority, and leadership. Participants repeatedly talked about two Yolŋu Elders, known as Joseph and Grant (pseudonyms as Joseph passed away and Grant was sick and unable to be reached to obtain consent to use his real name), who actively facilitated the preparedness and response activities through their engagement with disaster response agencies. For example, these Yolŋu Elders decided to use the loudspeaker from the Shire (Local Authority) office in the center of the community to update the community with Cyclone information in the local language every thirty minutes and to direct people to stay in the shelter until the Cyclones had passed by. Because of their great authority, agencies and Yolŋu strictly followed and adhered to the directions from the Elders during the Cyclones: 


*“Joseph was the most active guy. He was directing everyone, police, and security to carry out response and maintaining contact”*
(Dorothy).

However, participants informed the researchers that Joseph had died, Grant is sick, and no one has replaced them. This has created significant gaps in how the community communicates and engages with government agencies. The findings further suggest that the inability of the community to replace, develop, and empower the leadership in Galiwin’ku reflects how government persistence in applying colonial practices perpetuates community exclusion, disempowerment, and a lack of partnerships with the government. Without meaningful engagement, responsibility, and authority from the government, tribal leaders have lost their interests in local leadership. Valarie, a Traditional Owner, commented that


*“Our people and leaders are getting sick and tired. Because power is going this way and that way and it’s shrinking down our malay [clan] leaders.”*


The weak leadership practices imposed through colonial interference, in turn, reduced the availability and effectiveness of several core community capacities. Prominent here are social resources, such as communication, participation, inclusivity, collective efficacy, social capital, mutual trust, and partnerships with the government:


*“We had meetings and meetings, and nothing has improved, just waste of time. All the politics comes in the meetings, people say different things and don’t listen to others”*
(Rossmandi).

Redressing this issue also entails understanding other intrinsic influences on social functioning. The vital one here is the historical and contemporary influence of faith and spirituality.

#### 3.1.6. Yolŋu Having Faith in Supreme Power/God

Before colonization, Yolŋu believed in a supreme power that created all creatures and that is helping them. When the missionaries arrived, bringing Christianity, Yolŋu came to know God and the Bible. Yolŋu believe that the supreme power and God are similar and thus hold both their traditional beliefs in a supreme power and their belief in God. The importance of accommodating this in a DRR theory derives from the significant influence that Yolŋu’s spiritual/religious beliefs have on people’s understanding of the causes of natural phenomena and their potential to become natural disasters. Several participants referred to the Bible and expressed the belief that cyclones and other social disasters (e.g., polluting the seas) occur because Australia is a nation without a vision. This lack of vision leads to people perishing because they are not working together and not listening to the teachings of God. For example, Rosemary related the cyclones to the disaster mentioned in the Bible (Noah’s flooding) which eradicated evil from the society. Rosemary thought that cyclones were a warning from God to take the right path. These participants also believe that God is the only one that can keep them safe and that trusting that God will keep them safe provides people with confidence and strength:


*“When we were in the shelter and we all were talking to each other that if cyclone hits our community we have to pray together. People at Marthakal, school and shelter all were praying. I was little bit afraid, but I had a faith on God. I was thinking to forget everything and just pray to God. We know he cares for us and we had trust on him”*
(Djanice).

These spiritual/religious beliefs possess great relevance for promoting the values of morality, ethics, harmony, and the practice of collective efficacy.

The findings also show that having strong spiritual/religious beliefs helped to heal the damage to nature and contributed to mitigating the trauma caused by the Cyclones, to some extent. For example, Djandi mentioned the following:


*“When cyclone came, they were saying it would take 5 years for new leaves to grow but Yolŋu had faith upon God that he will heal things quickly”.*


However, while these beliefs provided individual-level psychological support, there was no consolidated mechanism for utilizing the potential of this belief system to diffuse the trauma at the community level, such as involvement of the Church for spiritual healing. Moreover, other cultural practices which are vital means of spiritual healing, such as sharing stories and experiences, were ignored in top-down government/agency management practices intended to address the trauma and stress created by the Cyclones.

The ensuing inability to address the trauma and stress post-Cyclones negatively influenced the community’s intrinsic relationships. People were unable to share, understand, and heal from the trauma produced by the Cyclones: *“People were scared, some just kept that to themselves being depressed”* (Galikali). A theme running through this and previous sections is how imposed organizational structures impede Yolŋu. A corollary of this is a need to develop an Indigenous organization that can manage social, spiritual, and environmental capacity building activities.

#### 3.1.7. Aboriginal-Controlled Organizations Developing Capacities

Aboriginal-led community organizations, such as the Yalu, Miwatj Health Aboriginal Corporation (Miwatj), and the Aboriginal Land Progress Aboriginal Corporation (ALPA), which are deeply embedded in the everyday lives of people across the region, represent resources for developing capacities that strengthen community DRR and support community development. Miwatj runs the Ngalkanbuy clinic that provides acute and preventive care programs in the community. Miwatj also maintains a list of vulnerable people in Galiwin’ku, which led to their priority evacuation during the Cyclones. The Miwatj building (health center), which can accommodate up to 100 people, has also been designated as an emergency shelter in disasters.

Yalu is a local research organization that partners with universities and organizations to design and implement community-driven research and health and wellbeing programs for the community. Both Miwatj and Yalu play an important role in providing health and wellbeing services at the grassroots level that are based on connections to kinship and country, and therefore, both are considered vital for sustaining Yolŋu’s strength and resilience.

ALPA is one of the biggest Australian Aboriginal organizations operating across the NT and Queensland. It was created to develop Indigenous enterprise, local economies, and businesses. In Galiwin’ku, ALPA runs two supermarkets and significantly contributes to employment generation in the community.

However, while these organizations contribute to supporting the health and wellbeing of the community, they do not include DRR functions (e.g., disaster preparedness or response training programs (see below)) in their remit. This reflects another legacy of the top-down disaster management process imposed by the (Territory) government Emergency Management Unit in Galiwin’ku (see below). It was through this research that Yalu realized both the constraints that top-down management created and the capacity of Yolŋu culture to make valuable contributions to DRR and the need for its inclusion in DRR planning through partnerships. To exemplify how Yolŋu culture can strengthen DRR, Yalu planned to weave the disaster facet into its next annual cultural festival: 


*“Thank you for this [research]. We should add this to our cultural festival somehow to show what our culture means to us to prevent disaster”*
(Tamara).

Taken together, the Yolŋu traditional culture and spiritual practices of strong connection to country; traditional knowledge; reciprocal sharing and caring; the role of Elders and women in preserving, exercising, and transferring ecological knowledge; and faith in spiritual forces emerged as the main strengths which contribute to DRR in Galiwin’ku. However, we found that the government mechanisms required to integrate these DRR strengths in enduring and sustainable ways, and in ways consistent with contemporary community development principles, are dysfunctional. This is primarily due to colonial approaches associated with Western governance models that impose general practices throughout the Northern Territory and irrespective of the inherent geographical and cultural diversity that exists within the jurisdiction. We will discuss these aspects in detail in the next section.

### 3.2. Colonizing Practices Weakening Yolŋu, Increasing the Risks of Disasters

Our study found that Yolŋu cultural capacities and practices that could be effectively applied to DRR, reducing the likelihood of extreme natural events becoming disasters, and being able to respond to disasters in timely and effective ways should disaster eventuate, have been and remain substantially undermined and weakened by the government imposing their Western perspectives and practices. The main colonizing strategy is the government’s top-down approach and its role in creating the contemporary inequities that Yolŋu are suffering from. In Figure 2, interactions illustrating how the impacts of colonization are perpetuating poverty, disease, violence, unemployment, limited access to education, overcrowding, and housing issues in the community, and as constraints on community DRR, are presented.

#### 3.2.1. Balanda Culture Is a Disaster for Yolŋu

Indigenous mental and physical health and well-being are dependent upon connections to kin and country, community life, and ancient governance systems (9). However, colonization caused the Yolŋu peoples’ loss of access to ancestral lands, territories, and knowledge, disrupting and eroding their cultural practices in the process. The ensuing dislocation from country and disconnection from traditional resources weakened the capacity of Yolŋu people, both physically and culturally, to deal with natural and man-made hazards.

Galiwin’ku was established in 1942 as a Methodist Mission. Missionaries forced Indigenous people from surrounding homelands to come and settle in the mission, for two main reasons: To convert them to Christianity and to alter the Indigenous lifestyle according to mainstream Western norms [57]. This forced dislocation interrupted the connections with the ancestral homelands, cultures, and practices, and brought illness and trauma to Yolŋu. However, people who were forced to move to Galiwin’ku could not continue living in the old ways. This was perceived as having contributed to the deteriorating health and social issues discussed above.

The separation from, and destruction of, ancestral lands and the erosion of traditional ways of living, including hunting and agricultural activities, also eliminated access to traditional healthy food. In parallel, the junk food introduced by the first settlers and now supermarkets characterized by a high saturated fat and sugar content (e.g., takeaways, soft drinks, and processed meat products) led to the Yolŋu diet becoming increasingly unhealthy and damaging people’s physical and emotional health. Consequently, there is a high prevalence of cardiovascular, renal, sensory, skin, and dental diseases in Galiwin’ku. Moreover, the high cost of junk food has also contributed to the widespread poverty observed in the community. Participants frequently referred to this imposed “culture” as the ‘real disaster’, with this being a source of all the issues they are experiencing, including dilution of the (traditional) resource base that could be available to support DRR. In multiple ways, the imposed colonial culture has thus substantially undermined and weakened the extensive capacities that traditionally prevailed in Yolŋu society, including those relating to their capacity to take ownership of their DRR beliefs and practices.

Missionaries and settlers forcing the settling of 16 clans that lived in semi-nomadic ways within different countries throughout the area and resettling them in one place also created a heterogeneous community that is prone to conflicts between clans. While the essence of the worldviews and practices the clans hold are the same, the nuances are diverse. They are the guardians of different song lines, are connected to different totems, and speak different languages. Although, over time, these clans have connected, mainly through intermarriages and thus kinship systems, in ways which have contributed to developing a shared overarching culture, different clans still retain strong identities based on their distinctive clan origins. This set-up created by colonizers, in turn, inhibits clans from communicating and cooperating because interaction can frequently result in conflicts between the clans: *“There are boundaries among Yolŋu clans and people stay in their clan or family groups”* (Tamara). Additionally, it was noted, *“That’s the problems. Clans get into fights over small things and don’t try to solve their problems. It’s been going around like that”* (Rossmandi).

#### 3.2.2. Government Walking on Top of Yolŋu

Participants frequently referred to the imposed Western systems as the *“government walking on top of Yolŋu,*” with this contributing to excluding and disempowering the community and suppressing Yolŋu life, culture, and capacities. A good example of the government “walking on top of” Yolŋu is evident in the emergency management planning and implementation governance systems and processes imposed on Galiwin’ku.

The local emergency response in Galiwin’ku is headed by the Local Controller. This role is fulfilled by the Balanda (Western) Police Officer-in-Charge. The Local Controller is appointed by the Territory Controller, who is the Balanda Police Commissioner of the Northern Territory. The Local Controller heads the Local Emergency Committee (LEC). The LEC consists of the representatives selected by the Territory and Local Controllers from the local Government and non-Indigenous non-government entities using Western governance processes, excluding Yolŋu Elders from the decision-making process. Although the presence of the LEC in the community helps craft and implement place-based emergency plans among relevant Balanda agencies, there are no Indigenous representatives from the community in the LEC to facilitate the inclusion of the community’s perspectives in designing and implementing disaster management plans.

Furthermore, the members of the Northern Territory Emergency Services (NTES) Volunteer Unit that operates in the community to assist the emergency management are all Balanda from different services in the community. No Yolŋu are included in this emergency volunteer unit. A consequence of this systemic exclusion of community involvement by Balanda agencies was that only one of the participants in this study knew about the local emergency plan—its location and content. The Local Controller being the Police Officer-in-Charge and the emergency plan being placed in the Galiwin’ku police station demonstrate why excluding Yolŋu undermines DRR—the police is widely distrusted by Yolŋu due to them being the enforcer of Western “justice,” including colonizing practices, such as removing Indigenous peoples from their homelands; controlling Indigenous lives; and propagating several harmful social policies, including the removal of children from the community (stolen generations) [58].

The narrow and static top-down disaster management practice applied has resulted in a reactive agency-driven approach in the community that has failed to both utilize and further strengthen the highly valuable diverse Yolŋu resources discussed above and to develop relevant individual and collective Western DRR capacities in ways that would support community-based DRR. Participants stated, for instance, that they have never been provided with any disaster preparedness or response training:


*“We also need trainings for emergency response. Only service people have these trainings but Yolŋu need the trainings as well how to prepare and how to remain safe”*
(Helen).

This lack of training, combined with exclusion from emergency management planning, has resulted in a poor awareness among participants about the Western emergency management processes the government has put in place for the community. This undermines the Yolŋu capacity to prepare and respond:


*“If [disasters] happens, I have to know, what is the process and how I have to organise and prepare”*
(Stephen).

However, an interesting finding was that during the immediate preparation for and responses to the Cyclones, the government agencies established partnerships with the community and drew upon local resources, such as Elders. Elders and the Local Authority, police, night patrol, clinic, and school worked together to spread warnings, preparedness actions, and response strategies in local languages. These partnerships demonstrated the value of including the community and integrating local capacities in DRR planning and implementation. Participants shared that their Western cyclone preparedness knowledge and practices, such as the need for decluttering unnecessary and heavy equipment ahead of cyclones, were obtained from these campaigns being delivered in local languages. Importantly, working in partnership with the government agencies helped develop the community’s connections with the government, which in turn enhanced the sense of empowerment, trust, responsibility, belonging, and reciprocity among Yolŋu: *“The way we worked in Burmulalals was very good. We worked in partnerships and together”* (Nyomba).

In contrast to the pattern emerging in the preparedness and response phases, in the recovery phase, government agencies reverted to the top-down approach, only involving the community in minor tasks (e.g., road clean-ups). A consequence of this exclusion was the emergence of several economic and social impediments to effective and timely community recovery, such as the failure to build adaptive capacity by training and employing Yolŋu to lead the recovery. All of the high-end and technical work related to infrastructure reconstruction and facility restoration was carried out by Balanda FIFO staff from other cities and states who did not know the community and who had to be flown in, accommodated, and fed at great expense (see below for details). The use of FIFO staff in these roles further reduces the likelihood of community development in the direction of greater and more sustainable community DRR and recovery knowledge and capability.

For future sustainable community development and DRR approaches, participants stressed the need for genuine partnerships between the government and the community as they occurred during cyclone preparedness and response phases. Participants believe that it is vital that such partnerships benefit from their cultural resources and for Yolŋu to be able to contribute to a coordinated effort towards DRR:


*“We need to be trained in emergency training, first aid, emergency response and look after and to support each other, instead of relying on Balanda all the time”*
(Rosemary).

Participants also reckon that it is vital to utilize and integrate Yolŋu and Western knowledge to develop a more comprehensive and effective DRR:


*“It’s not about learning everything (DRR) from balanda, it’s [ecological knowledges] already active inside Yolŋu and in lives of Yolŋu. We just need a bit of training”*
(Rosemary).

#### 3.2.3. Balanda Infiltrating Indigenous Governance Systems

According to the participants, Western top-down governance approaches have also infiltrated the governance of Indigenous organizations, such as the Local Authority (LA) governance, and contributed to a lack of communication and mistrust between the Yolŋu and LA. Galiwin’ku’s LA, locally known as “Shire”, is comprised of 14 elected representatives from all tribes. The Shire is supposed to work as a ‘bridge’ between the government and community members. However, participants perceived the Shire as inhibiting effective communication. This separation was especially apparent during the Cyclones:


*“What hurt us most during cyclone was not enough communication between Shire and the community”*
(Rosemary).

Participants also strongly questioned the contribution of the Shire to empowering and developing the community:


*“I never saw them [Shire] going out and stand for the rights of the people and support them”*
(Rossmandi).

Valarie, a Traditional Owner, and member of the Shire, stated that the executive management of the Shire comprises Balanda, who have the real power; Yolŋu representatives are powerless. Valarie continued that the Balanda executive management is always setting the agenda according to government priorities and thus controlling what gets discussed and excluding community voices: *“We don’t discuss these kind of community issues (disaster management) in the meeting because they [Balanda] got their own agenda, their own opinion”* (Valarie). It was through this research that Yolŋu started to understand and articulate the importance of Yolŋu involvement in disaster management planning and the need for this to be conducted through a bottom-up and community-driven approach. Valarie, as a Shire representative, decided to discuss this in the LA:


*“This is the first time I am going to raise and discuss this in the Shire meeting. It is through this research that we will write this proposal. We want to see young Yolŋu to be trained for emergencies and get jobs in emergency services”*
(Valarie).

The Western top-down governance style has even been imposed on community-controlled Aboriginal organizations; the Chief Executive Officers (CEO) of community organizations in Galiwin’ku, including Yalu, Miwatj, and ALPA, are Balanda. Although the Board of Directors of these organizations are Yolŋu, participants strongly emphasized that Yolŋu are only figureheads; the real power lies with the Balanda CEOs, who make and implement decisions that serve the best interests of the government and secure their jobs: *It’s balanda sitting there on top and bringing their own agenda”* (Rossmandi). Participants forcibly expressed the need to appoint Yolŋu CEOs in these organizations to facilitate power being returned to Yolŋu and to ensure true representation of local needs and perspectives in the programs conducted by these organizations. As Figure 2 illustrates, this kind of infiltration process is also influenced through the government’s emphasis on fly-in/fly-out workers.

#### 3.2.4. Fly-In/Fly-Out (FIFO) Workers Imposing Government Agendas

Participants viewed FIFO staff as the main mechanism for not only imposing the government’s agenda, but also inhibiting opportunities for developing skills, capabilities, and employment, thereby suppressing community development. For example, Nyomba referred to the FIFO approach as representing the short-term and narrow thinking approach of the government. She highlighted that FIFOs are only concerned about their jobs and cannot feel or know the deep impacts that disasters have on the people, such as spiritual trauma:


*“Balanda come here, work, get the Rupiah [money] and they fly out. They should have been here at the first place to see and experience the disasters to know what it is in real.”*


Maypilama discussed how FIFO staff do not inform, consult, and partner with the community to include peoples’ perspectives in the development of the services they provide: *“Mobs of Balanda suddenly appear and start doing jama [work] and community doesn’t know what’s going on”.* Helen highlighted that FIFO staff lack a knowledge of local culture and protocols, resulting in them interacting in culturally inappropriate ways with Yolŋu. This further increases mistrust and doubts in the community:


*“They [FIFO staff] go into the people’s property without permission and without taking anyone of us to come with them. It makes us uncomfortable, scared and embarrassed.”*


Stephen highlighted the exorbitant costs associated with using FIFO and how the FIFO approach hinders community development:


*“Half of the money goes for the airfares, accommodation and transporting the stuff from Darwin to here, instead of spending it on the community.”*


The FIFO issue has other implications, including its role in reducing meaningful employment opportunities for community members, as shown in Figure 2.

#### 3.2.5. Employment Opportunities Are Lacking in the Community

Galiwin’ku, like other remote Indigenous communities, has high levels of poverty. The findings suggest that poverty has multiple implications for DRR. For example, it increases the magnitude of the direct impacts that environmental and social hazards create in the community (e.g., inability to stockpile ahead of disaster warning). Poverty has additional indirect consequences, including those from limited access to health, education, and proper housing facilities.


*“Sometimes when there are warnings, people don’t buy food and emergency equipment like torch, candles and tin food because they don’t have Rupiah [money]”*
(Djandi).

While participants identified several reasons for poverty, they emphasized high unemployment as the main contributor, with the closure of fishing, farming, and timber businesses due to government regulation being prominent factors. These businesses were established by the missionaries in partnership with people and were the main sources of employment: *“We had fishing, farming, and timber industry here back in 70s early 60s and all the people were employed”* (Dorothy). Participants also emphasized the failure of the government to create sufficient job opportunities in the community for Yolŋu, especially for youth: *“You see the young boys and girls roaming on the road whole day and night because there are not enough jobs”* (Rosemary).

Specifically, participants brought up the concern that while several services operating in the community represent potential sources of employment (e.g., the school, police, and community development programs), most of these jobs are taken by non-Indigenous people, rather than Yolŋu being trained to do these jobs: *“We want to see more Yolŋu working in services than Balanda. At the moment there are more Balandas than Yolŋu”* (Nyomba). The disproportionate Yolŋu–Balanda employment ratio, particularly in the health, education, and social services sectors, has not only perpetuated economic disparity and thus conflict among Yolŋu who work and who do not work, but also created a significant gulf in trust between Yolŋu and the government.

To address unemployment issues, ALPA runs a Community Development Program (CDP). The CDP’s objectives are to train and create employment for Yolŋu in different fields, including furniture making, clothing manufacturing, fishery, and arts. However, CDP only provides the training and does not create long-term sustainable employment. Moreover, the kind of training provided is largely failing to develop long-term and sustainable employment skills and capacities:


*“CDP is doing the activities like making chairs and tables and mowing lawns instead of training people and providing proper jobs like emergency response, building, electricians”*
(Valarie).

Once short-term activities are over, attendees become jobless again; attendees are then compensated through (Centrelink) welfare payments instead of proper wages. Short-term activity-based employment and Centrelink payments hurt people’s sense of pride and ultimately disempower them, as the following account indicates: *“Centrelink Rupiah is for disabled people only. People getting pays for CDP work through Centrelink are not disable”* (Djandi).

#### 3.2.6. Western Technical and Higher Education Lacking in Community

The lack of technical and higher education facilities in Galiwinku to equip Yolŋu with the knowledge and skills required to secure employment in the Western system also surfaced as a major constraint in individual and community development contributing to unemployment, poverty, overcrowding, and diseases and thus to increasing the risks of disasters:


*“Kids are doing Year 12 here locally at Shepardson College or interstate but after that, they don’t have a facility for further training”*
(Stephen).

Galiwin’ku, being a remote Island community, is over a one-hour flight from the nearest city and a several hours flight away from major Australian cities. Therefore, to study beyond year 12, children need to leave the community and go to boarding schools, Technical and Further Education (TAFE), and universities in the major cities many flight hours away from Galiwinku. Recent research has shown that this needs to change [59]. These separations cause severe social and emotional health and wellbeing challenges among children and parents as Indigenous health and wellbeing is dependent upon connections to kin, community, and country [9,60]. Participants shared that, in mission days, technical education facilities in the community provided them with technical skills. These skills not only helped the community economically, by providing employment, but also by developing several other skills, including those that enabled the Yolŋu to develop ways of living and growing their own food and permitted people to build their own houses in ways that were more suitable for the climate. However, these activities ended with the decline of the missions.

#### 3.2.7. Overcrowded Housing and Culturally and Climatically Unsuitable Housing

Among the socio-economic susceptibilities, insufficient and inappropriate housing emerged as a vital factor that increases disaster risks for Yolŋu. A survey conducted in six Arnhem Land Indigenous communities, including Galiwin’ku, revealed that, on average, nine people are living in one house, but the authors suggest that this number is likely to be higher in reality [11,61]. While houses can be an important pillar of structural, social, and cultural protection against disasters, overcrowding undermines this asset in Galiwin’ku. Stephen’s account suggests that overcrowding makes it challenging to store things during disasters: *“We don’t have bigger and good houses with proper facilitates, like storage to put our stuff in”* (Stephen). Nyomba points to the structural vulnerability that insufficient housing creates:


*“We sleep in the bedrooms with all the suitcases and boxes and that’s dangerous. If burrmulala [cyclone] comes, they can fall and hurt us”*
(Nyomba).

Rossmandi brought up the wider social disadvantage that overcrowding creates for youth: *“When there is no space to stay or sleep at home, kids spend time on the streets, fighting and breaking in.”* Participants’ accounts indicate that overcrowding is a prominent predictor of other social issues in Galiwin’ku, including infectious disease risk, violence, conflict, unemployment, and school attendance, which is consistent with the literature [60,61].

After the Cyclones, the government promised to rebuild the destroyed and damaged houses in Galiwin’ku. The new cyclone-proof brick-structure houses add to the structural and psychological resilience of the people, compared to the previous wooden structures:


*“Nowadays Galiwin’ku is a safe place. The brick houses are safe. The old houses made of wood and sand were not very strong”*
(Djandi).

However, five years after the Cyclones, only 30% of the planned houses have been built, illustrating inadequacies in the execution of the post-Cyclone recovery and regeneration phases.

Furthermore, participants also shared that while brick houses are safer for cyclones, they get hotter in wet seasons compared to the wooden counterparts. The higher temperature inside the houses requires the higher use of air conditioning and thus leads to higher electricity costs, creating additional economic disadvantages [11,61]. Moreover, participants commented that the designs of the houses are agency-driven, and are unsuitable for the tropical climate and culturally inappropriate. The inappropriate and insufficient housing designs inhibit community social, cultural, and ceremonial gatherings and undermine kinships and relationships, which has contributed to weakening the cohesion among people and increasing the disaster risk: *“We need bigger veranda, so the family can sit around outside. But they are not listening”* (Stephen).

## 4. Discussion

Using a holistic and systematic approach, our study identified that DRR in Galiwin’ku is best defined as a complex socio-ecological system. The fact that all of the identified sub-categories, categories, and core categories cover different dimensions of the system (Figure 2) supports the contention that DRR is a complex and dynamic process that derives from the interaction of diverse personal and contextual (historical, environmental, spiritual, cultural, social, economic, and political) factors over time [25]. The performance of this system depends on the diverse interactions within the numerous interconnected sub-systems and it needs to be understood, developed, and applied accordingly.

As depicted in Figure 2, Yolŋu traditional knowledge and practices strengthen the people, reducing the risks that residents in Galiwin’ku face from natural hazards. Being deeply embedded in the everyday lives of Yolŋu, deep reciprocal relationships with country and ecological knowledge, strong reciprocal kinship relations, Elders’ wisdom and authority, women and men sharing power, and faith in a supreme power/God and Indigenous-led community organizations generate sustainable and long-term capacities to adapt to and deal with disasters. While developing and reiterating some of the claims made by earlier studies of Yolŋu cultural competencies for DRR [1,6,8,15,61], our study adds an in-depth historical and contemporary analysis of how these capacities interact and how they are influenced and utilized when they interact with other sub-systems in their ecology.

We found that traditional Indigenous DRR capacities are undermined by historical and contemporary colonizing practices. At the core of these colonial practices is the government’s top-down, agency-driven, and service-oriented approaches to disaster management. These approaches intersect with Yolŋu DRR capacities in ways that increase the risks of disasters in several ways. First, such approaches hinder the development of the self-driven, long-term, and sustainable DRR capacities of the community. Second, top-down activities fail to draw upon the community’s intrinsic socio-cultural capacities in planning and implementing disaster risk preparedness, reduction, response, recovery, and regeneration strategies. Third, these strategies fail to identify and understand the interactions between the underlying triggers of disaster risks from a community perspective, which has limited the ability of relevant agencies to address these triggers when engaging with the community in broader DRR and community development processes. Prominent among these triggers are the infiltration of Indigenous governance systems, usage of fly-in/fly-out workers, scarcity of employment, lack of technical and higher education, and housing being overcrowded and culturally and climatically unsuitable.

While the systemic nature of the above barriers to DRR makes it challenging for Yolŋu to address the barriers, there is still substantial potential for Yolŋu to transform many of these conditions to enhance their DRR through capacity building, increased agency, and greater self-organization [62]. Importantly, realizing this potential is largely related to reviving and strengthening the Yolŋu worldviews and cultural practices identified by this study. This finding is supported by the literature, which identified connection with the lands and seas; social networks; Indigenous knowledge, values, skills, and learning; local leadership and governance structures; contextual and culturally relevant infrastructure; and economic diversity as facilitating DRR [62,63].

Our grounded theory suggests that most of these characteristics exist in Galiwin’ku at the community level, but are not recognized, utilized, and integrated by non-Indigenous agencies. The lack of integration undermines the effectiveness of the immediate societal responses to natural disasters, as well as longer-term sustainable transformation and adaptation [25,64]. For example, strong relationships and bonds among people promote a collective culture, but are not included in an agency-driven community development process to promote the collective efficacy. Similarly, brick houses add to the structural resilience, but inappropriate designs hinder the ceremonial and other social gatherings vital for sustaining social bonding.

To integrate Yolŋu’s cultural capabilities and the capacities of non-Indigenous agencies of DRR identified in our research, we propose an Indigenous two-way partnership Community-Based Disaster Risk Reduction (CBDRR) theory (Figure 3). Central to our theory is that Yolŋu reclaim their power by reviving and strengthening Indigenous knowledge through two-way learning with non-Indigenous partners.

Community-based partnerships to engage with Indigenous people are advocated in most major development and DRR frameworks in Australia. Two-way partnerships can lead to the transformation of existing dysfunctional systems into new ways of doing things. Two-way partnerships create new opportunities for critically examining, questioning, and reflecting on stakeholders’ beliefs and actions, and facilitate engagement in dialogue to redefine problems and create potential solutions from different perspectives that facilitate undertaking and sustaining effective social actions [1,10,65]. Consequently, two-way partnerships can have a multiplier effect on Indigenous development.

First, two-way partnerships can act as a vehicle to empower Indigenous communities by including Indigenous peoples and their local needs, expectations, and perspectives in community development processes [36]. Second, such partnerships promote strength-based development by reviving and integrating the traditional practices and knowledge and creating new ways to enhance future capacities to deal with disasters, such as complementing cultural DRR practices and knowledge with Western modern capacities [64,65]. Third, they can facilitate a discourse that culminates in the comprehensive understanding of disaster risk adaptation and reduction from both Indigenous and Western knowledge for improving societal adaptive capacities [1,13,65]. Together, these outcomes develop the personal-, household-, and community-level capacities and capabilities which are fundamental to sustainable community-based DRR [25,64].

To support such transformations, our Indigenous two-way partnership CBDRR theory (Figure 3) proposes two areas of focus for local, state, and national governments. The first focus concerns the importance of developing local emergency management Indigenous governance structures [10,65]. The establishment of informed, customary, and effective bottom-up governance structures is paramount for including local people and their needs, perspectives, and priorities in DRR policy and planning [66]. Moreover, local participatory structures are more likely to be effective and sustainable in the long term compared to traditional top-down models, which may diminish after the immediate crises have passed [11]. For example, the partnership-based working structure established between Yolŋu Elders and government agencies during the Cyclone preparedness and response phases dissipated because of the government returning to applying top-down recovery processes, rather than sustaining community engagement and facilitating community empowerment.

The establishment of governance structures that support collaboration with Indigenous people across Australia is already yielding positive outcomes by increasing Indigenous involvement in natural hazard responses and benefitting from their ecological knowledge [63]. However, such developments are limited to fire management initiatives. Partnership governance systems need to be expanded to support the general preparedness and resilience of Indigenous communities against all types of hazards, including those that are both natural and man-made [63]. Currently, guidance for creating such structures is very limited in Northern Australia [61].

To create an effective Indigenous emergency management governance structure, there is a need for greater empowerment of Indigenous agency and leadership through local political governance structures, such as the Shire. As the Shire is comprised of leaders from all clans, it is more conversant with the local and traditional resources and opportunities required to create such structures. Local governance structures, in turn, could facilitate the identification and integration of Indigenous ecological and cultural adaptive capacities in DRR and community development [67]. At the agency level, local governance structures would need to partner with local emergency services in co-creating and co-implementing DRR activities identified by the community, including activities that are targeting DRR awareness and education, efficiently directing the allocation of resources, identifying channels for the community to contribute resources and disseminate information, identifying techniques to promote community participation, and sharing experiences with other communities for mutual learning [12].

The second area of focus involves policy formulation and implementation. Our theory attempts to offer pathways for transitioning from reactive, agency-driven, and stand-alone emergency service delivery to a more proactive, participatory, holistic, and integrated formulation, implementation, and evaluation of DRR. This emphasis of our theory advances the calls from others that strategies that integrate disaster risk management, community and economic development, and poverty eradication play a vital role in strengthening the transformation and adaptation of Indigenous communities [2,4,5,6,8,9,13,61,62]. Such integration can help to create employment, develop social capital, value local knowledge, promote a culture of disaster preparedness embedded in daily life, and strengthen community-agency and self-determination [1,22].

An integrated approach to DRR can be achieved through partnerships between government agencies, established local governance structures (as discussed above), and Indigenous community-led organizations (Figure 3). Indigenous community organizations are deeply rooted in the lives of people and have a better understanding of the processes and factors that impact Indigenous DRR [68]. These claims were validated by the performance of Yalu and Miwatj in the health and wellbeing sector in Galiwin’ku. With effective engagement with government and non-government agencies, these organizations have shifted their approach from delivery to participation. This participatory approach has transformed how health and wellbeing issues are being addressed at the community level. Bottom-up inputs facilitate understanding of how interdependency between people and their social and environmental context underpins the adaptive capacity, especially if linked to culturally appropriate intervention strategies [32]. Moreover, these organizations have provided a vital means of local employment, allowing for greater economic, physical, and psychological resilience.

However, as our study identified, they are unable to integrate the DRR facets in tandem with other community health and well-being activities due to centralized and agency-driven disaster risk management in Galiwin’ku. With a more decentralized and bottom-up approach, Indigenous community organizations can partner with government agencies in a two-way approach to provide culturally appropriate and sensitive perspectives regarding disaster risk perceptions, behaviors, and actions. Community organizations can play key roles in facilitating recovery by creating, harboring, and distributing social capital for collective actions [69]. Moreover, people are more likely to be motivated to participate in community actions if guided by their fellow community members through communal platforms [13]. These issues are currently minimally addressed due to the exclusion of soft components of resilience in government-led emergency planning.

The multisectoral collaboration model of Yalu and Miwatj could be replicated in ways that integrate DRR in addressing other identified socio-economic development issues, such as unemployment and housing. For example, the emergency management services could collaborate with ALPA and CDP to provide emergency training to local people and develop a community-based emergency response team. Such collaborations are vital not only for developing longer-term adaptive capacities, but also for replacing the FIFO approach with local resources, restoring trust between community and the government, and most importantly, helping to integrate local ecological knowledge with dominant Western DRR models. Similarly, to address financial shortcomings that pose challenges to community development [5,6], partnerships with community organizations could be developed to create culturally appropriate and ecologically sustainable employment opportunities. This will help to enhance the economic resilience, revive and preserve local knowledge, and promote a sense of ownership and empowerment in the community.

The DRR facilitators and constraints identified in our study, and the way in which they interact to influence disaster risk perception and actions/interactions of the community, may differ in other Indigenous communities due to highly contextual and heterogenous socio-economic, cultural, historical, and environmental Indigenous experiences, worldviews, and knowledge [70]. These differences make it imperative to conduct more local- and individual-level studies to develop DRR theories relevant to local and specific contexts. However, as an initial study aiming to holistically and systematically identify these factors and their mutual interaction from Indigenous peoples’ perspectives, our research may provide a template for future in-depth, local, and holistic community-based DRR research, not only in Australia, but worldwide.

## 5. Conclusions

There is an increasing emphasis in the DRR literature on a paradigm shift from reductive approaches to understanding complex socio-natural systems of disaster risk, towards more holistic and systemic approaches [21,25,29,55]. Such approaches are required to better understand the socio-cultural-environmental sources of risks that increase the likelihood of hazardous events becoming disasters and that identify opportunities for marshalling these resources to facilitate reducing the risks that arise from interactions among and between individual, social, economic, environmental, structural, and historical dimensions [25,32]. In response to this call, we conducted a holistic study with a very remote Indigenous community in the Top End of the NT in Australia.

Through our research, our grounded theory systematically identified that the multitude of Yolŋu cultural capacities, including relationships with the country and natural events, traditional knowledge, sharing and caring among people, the role of Elders and women, community organizations, and spiritual beliefs strengthen community DRR responses to manage disasters. However, these capacities are intersected and undermined by using top-down and agency-controlled Western emergency management approaches. The typical top-down governance model has resulted in community disempowerment through the systematic exclusion of Yolŋu needs, requirements, perspectives, and intrinsic strengths in DRR efforts, exacerbating the community’s disaster risks.

To address this issue, we propose, based on the grounded theory that emerged from the data, an Indigenous two-way partnership CBDRR theory that suggests intersectoral engagement between different DRR stakeholders at all levels, including local government, government DRR agencies, Shire, local governance structure of emergency management, and Indigenous community organizations. Intersectoral engagement through two-way CBDRR models can lead to Yolŋu reclaiming their power to develop and strengthen their capacities to support future disaster responses and sustainable Indigenous development by i) drawing upon traditional knowledge and practices that inform DRR in tandem with development policies, and ii) addressing socio-economic inequities from the community’s perspectives for greater agency and self-reliance [7,10,30,65].

## Figures and Tables

**Figure 1 ijerph-18-00855-f001:**
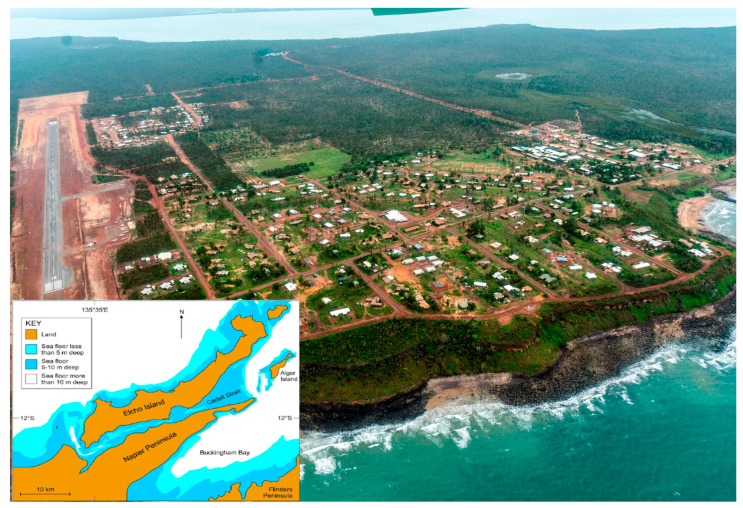
Galiwin’ku (source: https://www.eastarnhem.nt.gov.au/galiwinku).

**Figure 2 ijerph-18-00855-f002:**
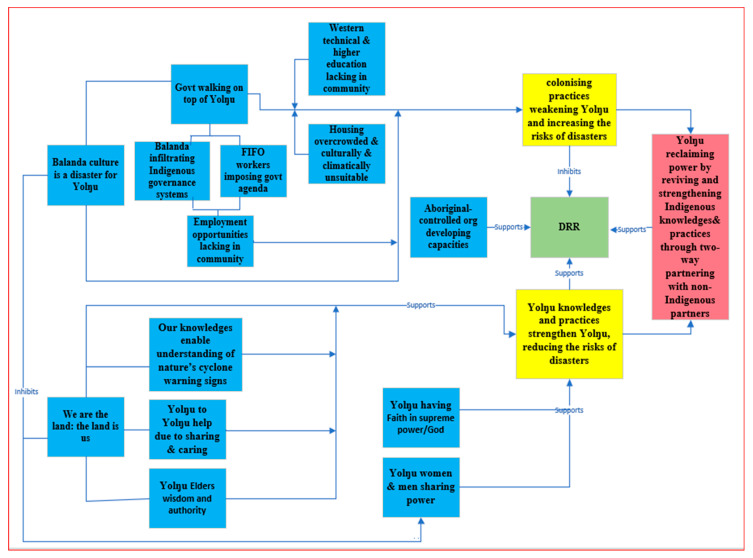
Grounded theory of Indigenous disaster risk reduction (DRR) in Galiwin’ku.

**Figure 3 ijerph-18-00855-f003:**
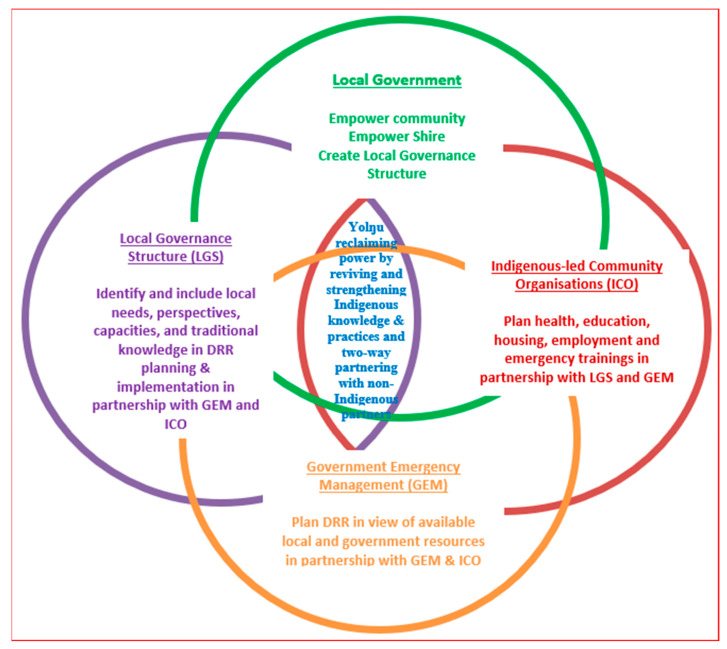
Indigenous two-way partnership Community-Based Disaster Risk Reduction (CBDRR) theory.

**Table 1 ijerph-18-00855-t001:** Details of data collection and participants.

Participation Details	No.	Participant Names
Total Participants	20	Community members: males
Males Females	5 15	Steven Stephen Shane Rossmandi Bobby Community members: females Djandi Dorothy Nyomba Rosemary Tamara Valarie Maypilama Glenda Djanice Margaret Sandra Tana Galikali Joan Raylene
Types of participation		
One-to-one conversations	10	
Yarning circle 1 Yarning circle 2 Yarning circle 3	3 4 3	
Duration of conversations		
One-to-one conversation		
*Maximum* *Minimum*	91 mins 30 mins	
Yarning circle 1 Yarning circle 2 Yarning circle 3	61 mins 26 mins 53 mins	

## Data Availability

The data has been stored in Charles Darwin University data repository as per University policy and is available upon request from the corresponding author.

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
