# Peer review of "Facilitating Sustainable Disaster Risk Reduction in Indigenous Communities: Reviving Indigenous Worldviews, Knowledge and Practices through Two-Way Partnering"

_ijerph, 2021, doi:10.3390/ijerph18030855_

Round 1

Reviewer 1 Report

This is a needed study in disaster sciences. The experiences of indigenous groups are often overlooked. I commend the authors for completing this study.

In table 1, all of the participants' name NEED to be removed. This is a violation of research ethics, and must be corrected before publication if these are the real names of the respondents. Having the names does not add anything to the paper. 

Again, in the findings section, the names of respondents should be removed to protect the identity of participants. The authors can use phrases such as "respondent 1" or "one female respondent indicated...", ect. However, the use of people's real names is a major ethical violation.  

I believe after the authors remove the names of respondents from the entire paper, this study is a major contribution to indigenous disaster studies. But, the names must be removed.

Author Response

In table 1, all of the participants' name NEED to be removed. This is a violation of research ethics and must be corrected before publication if these are the real names of the respondents. Having the names does not add anything to the paper.

Again, in the findings section, the names of respondents should be removed to protect the identity of participants. The authors can use phrases such as "respondent 1" or "one female respondent indicated...", ect. However, the use of people's real names is a major ethical violation. 

I believe after the authors remove the names of respondents from the entire paper, this study is a major contribution to indigenous disaster studies. But, the names must be removed.

Authors Response

We acknowledge the observation raised by the reviewer. However, an important aspect of decolonising Western research it to respect that some Indigenous participants consider it very important to be identified with their real names to show respect towards the relationships and to indicate their authority to share and present their information (Ashdown et al., 2018; Kovach, 2015; Kurtz, 2013). Our research was controlled and led by Indigenous supervisors, reference group and co-researchers and they decided to give participants in the consent form the option to choose for either their real names or pseudonyms being used when using their quotes in publications. Participants being able to choose their real names and using participants’ real names when using quotes from them in publications was approved by the Charles Darwin University and University of Canberra, Ethics Committees under clearance number H18060.

All the participants, except one, chose for their real names to be mentioned in the publications. Only one participant chose for the pseudonym and we have used this pseudonym to present her/his quotes in our manuscript. We have mentioned these aspects in the manuscript in lines 250-252 on page 7.

Accordingly, we suggest using the real names of the participants in our manuscript to uphold and promote the decolonising research and Indigenous data sovereignty (Kovach, 2015; Kurtz, 2013).

Reviewer 2 Report

Facilitating sustainable disaster risk reduction in an Australian

Indigenous community: Reclaiming power by reviving Indigenous knowledges

and practices through two-way partnerships

The title of the paper is too long. I suggest shortening it.  In summary, there is no information that the study concerns the island of Galiwin'ku (Elcho Island).The introduction is also too long. The purpose and test hypotheses should be given after a few sentences of the opening.Such a long introduction should be divided into two parts - the introduction and the literature review concerning reducing disaster risk. The literature review is crucial for researchers to explain a gap in existing studies.Please change the numbering of subpoints 1.1. Current Indigenous DRR research and practice, there is no 1.2. In the methodology section under Figure1: Galiwin'ku map (Source: https://northernterritory.com), it contains an active link, and the page displayed includes a tourist advertisement for the island of Galiwin'ku. Figure 1 is a map of the distance from Darwin to Galiwin'ku Island. I suggest you delete the figure or give the MAP OF THE ISLAND. 

The conclusions are exciting and well presented.

The technical side and the list of literature need correction.

Author Response

1.The title of the paper is too long. I suggest shortening it.

Authors Response: We have shortened the title. The new title reads as “Facilitating sustainable disaster risk reduction in Indigenous communities: Reviving Indigenous worldviews, knowledges and practices through two-way partnering”.

2. In summary, there is no information that the study concerns the island of Galiwin'ku (Elcho Island).

Authors Response: We have added “Galiwin’ku” in the summary in line 23.

3. The introduction is also too long. The purpose and test hypotheses should be given after a few sentences of the opening. Such a long introduction should be divided into two parts - the introduction and the literature review concerning reducing disaster risk. The literature review is crucial for researchers to explain a gap in existing studies.

Authors Response: We have divided the introduction into 2 parts as the Reviewer suggested. The purpose and research questions have been mentioned in 3rd paragraph, lines 66-78 under the heading “Introduction”. The sub-section 1.1 “Disaster risks of the North Australian remote Indigenous communities” overviews the literature regarding the disaster risks faced by Indigenous communities of Northern Australia. The sub-section 1.2 “Current Indigenous DRR research and practice” reviews the current Indigenous research and policy formulation and implementation for reducing risks in Northern Australia and gaps in existing discourse.    

4. Please change the numbering of subpoints 1.1. Current Indigenous DRR research and practice, there is no 1.2.

Authors Response: There are now sub-sections 1.1 and 1.2 in the manuscript.

5. In the methodology section under Figure1: Galiwin'ku map (Source: https://northernterritory.com), it contains an active link, and the page displayed includes a tourist advertisement for the island of Galiwin'ku. Figure 1 is a map of the distance from Darwin to Galiwin'ku Island. I suggest you delete the figure or give the MAP OF THE ISLAND.

Authors Response: We have added Galiwin’ku community’s map on page 6 showing the community as well as geographical position of the community.

6. The conclusions are exciting and well presented.

Authors Response: We thank the Reviewer for the comments.

7. The technical side and the list of literature need correction.

Authors Response: We have reworked and corrected the list of literature cited, and other grammatical and typo errors.

Acknowledgement:

We thank the Reviewer for the feedback to revise and resubmit our paper, which we believe makes now a much stronger contribution to the relevant literature.

Round 2

Reviewer 1 Report

The author's response to my previous comment is accepted and I see where they inserted the information about respondents indicating they wanted to used their names. However, in addition to the sentence they add, the author should include the following information in the same area of the paper that they used to justify their methods. Others will find issue for the same reasons I described previously, but if the following description is provided in the paper, it will help to educated them: 

An important aspect of decolonising Western research it to respect that some Indigenous participants consider it very important to be identified with their real names to show respect towards the relationships and to indicate their authority to share and present their information (Ashdown et al., 2018; Kovach, 2015; Kurtz, 2013). Our research was controlled and led by Indigenous supervisors, reference group and co-researchers and they decided to give participants in the consent form the option to choose for either their real names or pseudonyms being used when using their quotes in publications. Participants being able to choose their real names and using participants’ real names when using quotes from them in publications was approved by the Charles Darwin University and University of Canberra, Ethics Committees

Author Response

Reviewer's comment: The author's response to my previous comment is accepted and I see where they inserted the information about respondents indicating they wanted to used their names. However, in addition to the sentence they add, the author should include the following information in the same area of the paper that they used to justify their methods. Others will find issue for the same reasons I described previously, but if the following description is provided in the paper, it will help to educated them.

Authors' response: We thank the reviewer for this important feedback. Agreeing to the reviewer's comment, we have added the required information on page 7, lines 247-256. We have also proofread the paper for minor spell check as the reviewer highlighted.